



# 1 Calibration of a photoacoustic spectrometer cell using light
# 2 absorbing aerosols. A technical note.

Nir Bluvshtein,[1] J. Michel Flores,[1] Quanfu He,[1] Enrico Segre,[2] Lior Segev,[1] Nina Hong,[3] Andrea
Donohue,[3] James N. Hilfiker[3] and Yinon Rudich[1,*]
[1] Department of Earth and Planetary Sciences, Weizmann Institute of Science, Rehovot 76100, Israel
[2] Physical services, Weizmann Institute of Science, Rehovot 76100, Israel
[3] J.A. Woollam Co., Inc., 645 M Street, Suite 102 Lincoln, NE 68508, U.S.A.
*Correspondence to:* Yinon Rudich (yinon.rudich@weizmann.ac.il)

## 11 Abstract

Multi-pass photoacoustic spectrometer (PAS) is an important tool for the direct measurement of
light absorption by atmospheric aerosol. And accurate PAS measurements require an accurate
calibration of its signal. Ozone is often used for calibrating the PAS by relating the instrument
signal to the absorption coefficient measured by an independent method such as cavity ring down
spectroscopy (CRD), cavity enhanced spectroscopy (CES) or an ozone monitor. We report here a
calibration method that uses measured absorption coefficients of aerosolized, light absorbing
organic materials and offer an alternative approach to calibrate the PAS at 404 nm. To
implement this method we first determined the complex refractive index of an organic dye using
spectroscopic ellipsometry, and then we use this well characterized material as a standard
material for PAS calibration.

## 23 Introduction

24       Light absorption by atmospheric aerosols still poses one of the greatest uncertainties

associated with the effective radiative forcing due to aerosol-radiation interactions (IPCC, 2013).
Absorption of incoming solar radiation exerts positive radiative forcing to the top of the
atmosphere due to heat transfer from light absorbing aerosols to their surroundings. It may also
lead to stagnation and clouds dissipation. In terms of positive radiative forcing, black carbon
(BC) aerosols are commonly considered to be second only to $CO_2$ (Bond et al., 2013) with strong
absorption throughout the solar spectrum. Characterization and quantification of light absorbing
organic aerosols, referred to as brown carbon (BrC), have been given increasing attention in the
past decade (Andreae and Gelencser, 2006; Bond and Bergstrom, 2006; Alexander et al., 2008;
Flores et al., 2012b; Lack et al., 2012a; Flores et al., 2014; Laskin et al., 2015; Moise et al.,
2015). Although the atmospheric burden of BrC is estimated to be more than three times that of
BC (Feng et al., 2013) its absorption is strongly spectral-dependent with strong absorption in the
UV and visible spectrum and weak to non-absorbing in the longer wavelengths (Hoffer et al.,
2004; Kirchstetter et al., 2004; Kaskaoutis et al., 2007; Sun et al., 2007; Chen and Bond, 2010;
Moosmuller et al., 2011; Lack et al., 2012a).
Light absorption properties of BrC and its mixing with absorbing and non-absorbing aerosol
components introduce a need for sensitive and accurate direct measurement of light absorbing
aerosols which is still very challenging. For example, filter based techniques such as the Particle
Soot Absorption Photometer (PSAP), the Multi Angle Absorption Photometer (MAAP) and the
aethalometer, require correction factors which are based on some *a-priori* information regarding
aerosol type and source and have accuracy in the range of 20% to 35% (Bond et al., 1999;
Weingartner et al., 2003; Collaud Coen et al., 2010; Muller et al., 2011).
Multi-pass photoacoustic spectrometer (PAS) at several wavelengths throughout the visible
spectrum has the potential to produce sensitive direct measurements of light absorption due to
BrC aerosols. Coupling PAS instruments to a thermal denuder and cavity ring down
spectrometers (CRD-S) to measures the absorption and extinction coefficients simultaneously at
temperatures ranging from ambient to over 450 $^{0}$C allow attribution of aerosol light absorption to
BrC, BC, and BC with enhanced absorption due to less-absorbing coating (Cappa et al., 2012;
Lack et al., 2012b).
In a PAS cell, a modulated laser light is absorbed by a sample of particles or gas, generating
a modulated acoustic wave with intensity that is proportional to the energy absorbed by the
sample. This acoustic wave, which is detected by a sensitive microphone, has a characteristic
radial and longitudinal resonance when the light source is modulated at the cavity resonance
frequency ($F_r$). A more detailed description of the PAS method for aerosol light absorption
measurement may be found in Arnott et al., (1999) and Nagele and Sigrist, (2000).
While direct, *in-situ* aerosol absorption measurement using PAS avoids the disadvantages of
filter based techniques, an accurate calibration procedure is required to relate the instrument




signal to the absorption coefficient ($\alpha_{abs}$). One way of achieving this is to directly relate the $\alpha_{abs}$
to the microphone response and the laser power by a theoretical relation (Arnott et al., 1999):

$$\alpha_{abs} = \frac{P_{mic}}{P_{Laser}} \frac{A_{res}}{\gamma - 1} \frac{\pi^2 F_r}{Q} \qquad (1)$$

$$Q = \frac{F_r}{FWHM} \qquad (2)$$

where $P_{Laser}$ is absolute laser power in the resonator, $A_{res}$ is resonator cross sectional area, $\gamma$ is
isobaric to isochoric specific heats taken as a constant in dry air, $P_{mic}$ is the microphone signal
power, Q is the resonant cavity quality factor and FWHM is the full width half maximum of the
acoustic response curve. The $F_r$ and the FWHM are sensitive to temperature, pressure and type of
carrier gas.

68          When the laser intensity inside the PAS cell is unknown or when it is not possible to

measure, as in the case of an astigmatic cell alignment, the instrument's response needs to be
calibrated empirically. This involves comparing the PAS signal to an independent transmittance
measurement where scattering is negligible (Arnott et al., 2000; Lack et al., 2006). A common
PAS calibration procedure is done by comparing direct measurements of $\alpha_{abs}$ at several
concentrations of an absorbing gas by cavity ring down spectrometer (CRD-S) with a parallel
measurements by the PAS cell (Lack et al., 2006; Lack et al., 2012b; Lambe et al., 2013). This
method is applicable when the Rayleigh scattering by the gas molecules is several orders of
magnitude smaller than the absorption and can be neglected. Since gas absorption cross sections
can be highly dependent on the wavelength (Vandaele et al., 2002; Bogumil et al., 2003), it is
essential to precisely know the wavelength of the light source used in each instrument or to use
the same light source in both instruments. At $\lambda = 404$ nm wavelength, $NO_{2(g)}$ is highly absorbing
with molecular absorption cross section ($\sigma_{abs}$) of $6.12 \times 10^{-19}$ cm$^2$ molecule$^{-1}$ (Bogumil et al.,
2003). However, at this wavelength $NO_2$ has a large quantum yield ($\Phi$) of 0.44 (Troe, 2000) and
it readily photolyases. Without accurate determination of the laser's power, it is difficult to
quantify the photolysis. At the same wavelength, $O_3$ has the advantage of being stable with $\Phi$
approaching zero (Bauer et al., 2000) and it is easily produced on-site or in the laboratory.  The
disadvantage of $O_3$ for PAS calibration at $\lambda=404$ nm, is that its $\sigma_{abs}$ is about 4 orders of
magnitude lower than that of $NO_2$. Different studies reported a wide range of $\sigma_{abs}$ for $O_3$ at $\lambda =$



404 nm ranging from $1.5 \times 10^{-23}$ to $6.3 \times 10^{-23}$ $cm^2$ molecule$^{-1}$ (Burrows et al., 1999; Voigt et al.,
2001; Bogumil et al., 2003; Axson et al., 2011) (Figure 1). For this reason, $O_3$ calibration
requires very high concentration (in the order of 100's to 1000's ppmv) which may cause
equipment degradation. An additional concern is that at concentrations in the order of 1000's
ppmv, $O_3$ may change the $F_r$ of the PAS cell. The extent of this effect depends on the $O_3$
concentration and on the instruments' sensitivity to gas composition i.e. the Q, and it can also be
easily calculated using a simple thermodynamic model for the speed of sound. In such a case the
laser modulation frequency should be adjusted to the new $F_r$ value.
An alternative calibration method is to use a standard aerosol with well-known absorption
properties. PAS calibration using size selected light absorbing particles requires a standard
material with accurate information of its complex refractive index at the instrument's
wavelength, which is not widely available. This procedure is also time consuming in comparison
to the use of a light absorbing gas and may be more difficult to implement on field and aircraft
applications.
Lack et al., (2012b), reported the development of the current PAS instrument. They
calibrated their PA-CRD-S (PAS coupled to a CRD-S) cells at 405 nm and at 532 nm with $O_3$
and commented that $NO_2$ calibration at 405 nm is possible using a photolysis correction factor
for the CRD-S measurements. Several other publications used the same instruments as in Lack et
al., (2012b) using $O_3$ calibration procedure (Cappa et al., 2012; Lack et al., 2012a; Lambe et al.,

106    2013).

Using a similar PAS cell as Lack et al., (2012b), we attempted to measure $\alpha_{abs}$ and
extinction coefficients ($\alpha_{ext}$) of brown carbon (BrC) proxy materials using the PA-CRD-S
following calibration of the PAS using $O_3$. The results yielded very high $\alpha_{abs}$ values which were
not consistent with other measurements. Therefore, we developed a reliable procedure to
calibrate the PAS instrument using light absorbing particles produced in the laboratory with a
widely available water soluble absorbing material – nigrosin. In this study, we describe the
details of this procedure which includes high accuracy measurement of the nigrosin complex
refractive index (RI) using spectroscopic ellipsometry. We also show that there are significant
differences between the PAS response curve calculated using nigrosin particles and the PAS
response curve calculated using $O_3$.





## Methodology

### Photoacoustic aerosol spectrometer

The multi-pass astigmatic PAS cell that is used in this work is described in Lack et al., (2012b) and only a brief description is given here. It is composed of dual half-wavelength resonators (11 cm long, 1.9 cm diameter) capped on either end with 1/4 wavelength acoustic notches. The total sample cell volume is 185 cm$^3$. While both resonators are open to the sample flow, only one is exposed to the modulated laser light; the other is planned for noise cancellation. Microphones are placed at the antinode of the sound wave in the center of each resonator and the speaker is placed at the background resonator. The $F_r$ of the system is found by producing 1 sec segments of white noise using the speaker located in the reference resonator. Each segment is sampled by the microphones at a 100 kHz rate and the $F_r$ is found by performing a Fast Fourier Transform. Examples of power spectra with different carrier gas compositions are shown in Figure 2 were $F_r$ is the frequency at the peak of the fitted Lorentzian curve. Typical Fr and Q values for our instrument are in the range of 1360–1385 Hz and 40-50 (unitless), respectively, over the pressure (97–101 kPa), relative humidity (RH; 0 - 11% RH) and temperature (20 to 24 °C) ranges, and with two carrier gases ($N_2$ or synthetic air). The instrument described in Lack et al., (2012b) produced an $F_r$ in the range of 1320 to 1360 Hz, Q in the range of 50 to 90 over pressure and temperature ranges of 20 to 90 kPa and 12 to 23 °C when dried air was used as carrier gas.

The astigmatic optical configuration consists of two high reflectivity mirrors (ARW Optical, Wilmington, NC, USA; dielectric coating R > 99.5%), 1.5" diameter, spaced 35 cm apart. The laser side mirror has a cylindrical radius of curvature of 43 cm and a 2 mm hole drilled in the center. The back mirror has a cylindrical radius of curvature of 47 cm, and is rotated 90° to the radius of curvature of the laser side mirror. Astigmatic alignment is achieved by aligning the laser through the 2 mm hole drilled in the center of the first mirror and on to an off-center target on the second mirror. Each following reflection is also directed to an off-center target on the opposite mirror. Each mirror was placed on kinematic mirror mounts for easy alignment (KM200, with an AD2-1.5 adaptor; Thorlabs, U.S.A). The PAS cell is mounted within the path of the laser multi-pass and is covered by 50 mm thick acoustic foam. The laser light passes





through the PAS cell through two 1 mm thick windows (CVI Laser, Albuquerque, NM, USA),
each with a high transmissivity (T > 99.5%) antireflective coating.

**Cavity ring down spectrometer**

A detailed description of the CRD-S method for aerosol light extinction measurement may be
found in Pettersson et al., (2004), Abo Riziq et al., (2007), Smith and Atkinson, (2001),
Bluvshtein et al., (2012) and references therein. The CRD-S used in this study differs from the
one described in a previous publication (Bluvshtein et al., 2012). Here the laser modulation rate
is varied to meet the PAS cell $F_r$. The cavity length and the aerosol filled length were extended to
95 and 80 cm respectively. Additionally, the gas/aerosol inlet to the cavity was moved to the
center of the cavity with two outlets at the cavity sides (Figure 3) from which the gas/aerosols
are pulled out. In this configuration the uncertainty associated with the ratio of cavity length to
aerosol filled length is reduced significantly regardless of flow conditions (i.e. ratio of sample
flow to mirror purge flow and cavity inner diameter). Discussion on errors associated with $R_L$
uncertainty may be found in Miles et al., (2011) and Toole et al., (2013).

**Photoacoustic aerosol spectrometer coupled to a cavity ring down aerosol spectrometer**

The photoacoustic aerosol spectrometer coupled to a cavity ring down aerosol spectrometer
(PA-CRD-S) described in this section (Figure 3) is composed of a 110 mW 404 nm diode laser
(iPulse, Toptica Photonics, Germany) modulated in the measured PAS resonance frequency at
50% duty cycle. The laser beam is split into two separate optical paths (CRD-S and PAS) using a
variable polarized beam splitter. The beam splitter is composed of a quarter waveplate (¼λ) and
a polarizing beam splitter (PBS). With the current setup, turning the ¼λ between 0 and 90° varies
the intensity ratio between the two optical paths from 0:1 to 1:1 CRD-S to PAS optical path,
respectively. The beam directed to the PAS is turned and aligned into the PAS cell through a set
of two plano-convex lenses (focal lengths of 30 mm and 50 mm) used as a telescope in order to
collimate the beam into a diameter of about 1.5 mm. The beam, directed to the CRD-S, passes
through another ¼λ plate, which together with the PBS serves as a variable attenuator protecting
the laser head from the beam reflected backwards by the CRD-S highly reflective mirror. This
back-reflected beam (dashed arrow in Figure 3) is transmitted through the PBS and is detected
by a photodiode and is used as an external trigger source for the CRD-S intensity decay
measurement. The forward beam is then turned and aligned into the CRD-S cavity by a set of



turning mirrors. While the PAS sensitivity depends on the laser power, the CRD-S system
requires only the minimal laser power needed by the photodiode. This allowed us to divert
approximately 78% of the laser power (about 86 mW) to the PAS cell and thus optimize its
sensitivity.
**PAS calibration**
To calibration the PAS cell, gas flow was pulled and split between the PAS and the CRD-
S at a flow ratio of 3:1 to equal the volume ratio of the two instruments. $O_3$ was generated by a
constant flow of high purity (99.999%) $O_2$ through a UV lamp $O_3$ generator (model 300, Jelight
Company, Inc. CA, U.S.A) for up to 800 ppm $O_3$ and through a corona discharge ozone
generator (model L21, Pacific Ozone, CA, U.S.A) for up to 4000 ppm $O_3$. The $O_3$ out flow was
mixed with dry $N_2$ to a 90% $N_2$ 10% $O_2/O_3$ mixture. The $O_3$ concentrations were varied by
adjusting the height of the cover glass of the UV lamp $O_3$ generator and by adjusting the voltage
gauge of the corona discharge ozone generator. At each gas concentration, data were acquired at
a rate of 1 Hz and averaged over intervals of two minutes.
**Measurement of the complex refractive index of nigrosin by Spectroscopic ellipsometry**
To infer the complex refractive index (RI) of nigrosin at λ = 404 nm from ellipsometry
measurements, five silicon wafers (surface area - 4 cm$^2$) with 300 nm of silicon thermal oxide
(Virginia Semiconductor, VA, U.S.A) were coated with nigrosine using concentrated aqueous
solutions and a spin coater (WS-400A-6NPP/LITE; Laurell Technologies Corporation, PA,
U.S.A). The concentration of the nigrosin solution was 1.25 and 1.5 times the room temperature
solubility limit of nigrosin (10 gr L$^{-1}$) and was kept at 40°C under constant stirring to maintain
solubility. Spin coating was done in two stages, a coating stage and a drying stage. During the
coating stage each sample was covered by the nigrosin solution and the spinning was done at
100, 500 or 700 RPM under dry $N_2$ flow for 14 minutes. The drying stage was performed at 3500
rpm for 1 minute in order to reach complete dryness and to remove liquid droplets adhering to
the wafer edges. Dry $N_2$ flowed from below the wafer stage in an upward direction so it will not
affect the liquid spreading on the wafer.
Spectroscopic ellipsometry is a proven method to determine thin film thickness and
complex refractive index (m = $n$ + i$k$) of materials (Fujiwara, 2007). Briefly, ellipsometry uses





polarized light to characterize thin film and bulk materials. A change in the electric field
amplitude and phase for p- and s- polarizations is measured after reflecting light from the
surface. Thin film thickness and optical constants ($n$ and $k$) are derived from the measurement.
Spectroscopic ellipsometry measurements were performed on the five film samples (described
above) using a J.A. Woollam M-2000 DI ellipsometer in the spectral range of 193 nm to 1700
nm at angles of incidence of 55°, 65°, and 75°. The instrument was pre-calibrated with a
calibration wafer to minimize systematic errors that are related to angle, wavelength and delta
offsets. In case of light absorbing materials $k$ is often correlated with the sample thickness. To
overcome this issue, we employed the interference enhancement technique to improve sensitivity
to light absorption as described in Hilfiker et al., (2008). The resulting film optical constants
were evaluated by comparison with the optical constants obtained from a simultaneous analysis
of all five samples (multi-sample analysis; MSA). With the two methods, we obtained high
sensitivity to light absorption. The Kramers-Kronig consistent complex refractive index of the
nigrosin films was modeled using five Gaussian oscillators along with a Sellmeier function. The
best-fit model was determined by the Levenberg-Marquardt regression algorithm and tested for
both statistical errors and model systematic errors. Statistical errors were estimated by the
Bootstrap re-sampling method (Rosa, 1988) and the model systematic errors were estimated
using the difference between the measured data and the best-fit model generated data.

**PAS calibration with measurements of nigrosin aerosol**

A nigrosin solution was atomized, and the resulting aerosol dried, size selected (250 nm
to 325 nm at 25 nm steps) (Bluvshtein et al., 2012; Flores et al., 2012a) and the absorption signal
was measured with the PAS instrument at several number concentrations (counted by a
condensation particle counter; CPC). Size selection was performed using an electrostatic
classifier (3080L, TSI, MN, U.S.A) equipped with an impactor (nozzle diameter of 457 µm).
Sample flow was set between 1 to 0.7 LPM such that the 50% cut-off diameter of the impactor
was 50 nm above the selected size. The impactor was used to reduce multiply charged particles
contribution. The signal of the PAS was compared to the aerosol $\alpha_{abs}$ calculated using Mie theory
algorithm from the complex RI retrieved from the dry film SE measurements together with the
particles number concentration.



## Results

Figure 4 shows the result of a single sample and a multi-sample analysis of the spectroscopic
ellipsometry complex RI retrieval, at 300 nm to 800 nm range. The single sample analysis shown
was performed on the sample with the thickest retrieved nigrosin film (137.2 ± 0.3 nm, coated
with 15 gr L$^{-1}$ nigrosin solution at 100 RPM). The imaginary part from the SE analysis is in good
agreement with the imaginary part calculated from aqueous solution UV-Vis absorption
measurement (Sun et al., 2007). The density of nigrosin for this calculation was taken as 1.6 gr
cm$^{-3}$ (Moteki et al., 2010). The complex RI of nigrosin at $\lambda$ = 404 nm was determined by the
spectroscopic ellipsometry analysis to be m = 1.624 (±0.008) + $i$ 0.154 (±0.008). The summed
precision and accuracy of the retrieved complex RI are about 0.5% for $n$ and 5% for $k$. Figure 4
also shows previously published complex RI values for nigrosin retrieved at 532 nm and 355 nm
wavelengths using CRD-S (Lack et al., 2006; Dinar et al., 2008; Lang-Yona et al., 2009;
Bluvshtein et al., 2012; Flores et al., 2012a). Such wide spread of complex RI values emphasizes
the need for a more accurate measurement for future use of nigrosin as a standard material, and
the limitations of the CRD method, that can benefit from a new well-established standard.
To further verify the validity of the new calibration approach we have used Pahokee peat
fulvic acid (PPFA) and Suwannee river fulvic acid (SRFA) which are often used as a proxy
material for atmospheric brown carbon due to their complex organic composition and their UV-
Vis absorption spectrum. In an accompanying paper Bluvshtein et al., showed that the mass
absorption cross section (MAC) of PPFA and SRFA, calculated from UV-Vis aqueous solution
absorption spectrum, is within the value range of the MAC calculated for ambient water soluble
organic aerosol collected during a biomass burning event.
Size selected PPFA and SRFA particles were measured with the PA-CRD-S and the complex
RI was retrieved from the CRD-S measurements using a Mie theory algorithm taking into
account the multiply charged particles (MCP) contribution (Flores et al., 2012a; Washenfelder et
al., 2013; Bluvshtein et al., 2016). The imaginary part of the complex RI was also calculated
from UV-Vis aqueous solution absorption measurement using material density estimation of 1.1
to 1.3 gr cm$^{-3}$. Our best estimation of the complex RI of PPFA and SRFA at $\lambda$ = 404 nm are m =
1.699 (±0.012) + $i$ 0.036 (±0.010) and m = 1.685 (±0.020) + $i$ 0.013 (±0.010) respectively. This
information together with the measured particle number concentration and MCP contribution



was used to calculate $\alpha_{abs}$ using Mie theory. Calculated $\alpha_{abs}$ of PPFA and SRFA are plotted
against the measured PAS signal in Figure 5. In addition, Figure 5 shows an $O_3$ calibration curve
with a slope of $4.975 \times 10^{-7}$ cm$^{-1}$ V$^{-1}$ and a nigrosin calibration curve with a slope of $2.533 \times 10^{-7}$
cm$^{-1}$ V$^{-1}$. Figure 5 clearly demonstrates that the PAS response curves calculated for the three
types of organic aerosols agree with each other, while the slope of the response curve produced
with $O_3$ over-estimates the instrument's response by a factor of about two. This implies that
measurements of aerosols $\alpha_{abs}$ at $\lambda = 404$ using PAS calibrated with $O_3$ may be significantly over
estimated.
With a parallel flow configuration, higher loss of $O_{3(g)}$ molecules to the PAS (aluminum)
walls in comparison to the CRD-S walls (stainless steel and lower surface to volume ratio) would
result in an underestimation of the PAS response and an overestimation of the calibration slope.
A similar artifact could result from reaction of the $O_{3(g)}$ with residual aerosol material on the
CRD-S walls, producing ultra-fine light scattering particles. These particles, if produced would
increase the CRD-S $\alpha_{ext}$ and cause an overestimation of the calibration slope. Repetitions of the
calibration procedure in tandem flow configuration, linearity and repeatability of the calibration
curve and the stability of the CRD-S signal ruled-out these affects as possible causes for the
overestimation of the PAS response due to the $O_{3(g)}$ calibration procedure.
Additionally, we did not find any literature information regarding significant energy
relaxation processes following UV-Vis light absorption by $O_{3(g)}$ which do not involve thermal
conversion.
**Conclusion**
In this study we demonstrate a new calibration for PAP instrument using nigroisn, a widely
available water-soluble absorbing material. We have derived the complex refractive index of
nigrosin throughout the UV and visible range using spectroscopy ellipsometry and suggest that it
can now be used as a standard material to calibrate PAS instruments at the UV-Vis-NIR
wavelength range for measurements of light absorbing aerosols. Nigrosin can also be used to
validate other chosen PAS calibration procedures. Our measurements also imply that calibration
of PAS with $O_3$ at 404 nm may lead to over-estimation of light absorption by aerosol.





As shown in this study, spectroscopic ellipsometry may be used to accurately determine the
complex RI of other organic dyes that may be used for the same purpose. It requires, however,
the production of uniform films of the studied material.
**Figures**

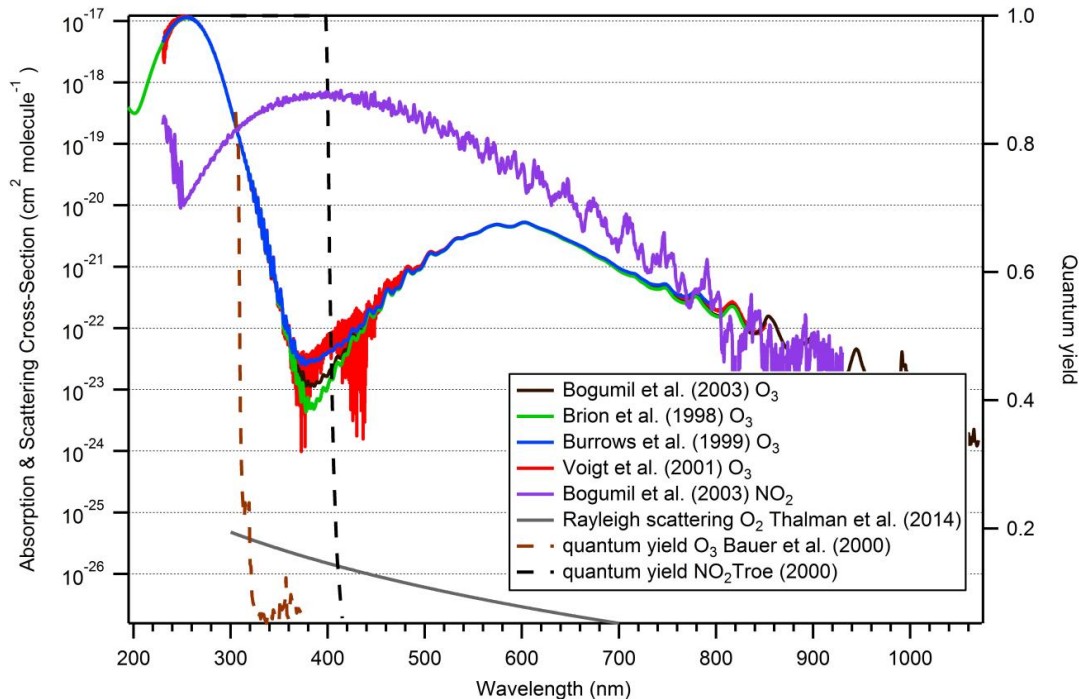


Figure 1: Spectral absorption cross-sections and quantum yields of $O_3$ and $NO_2$.





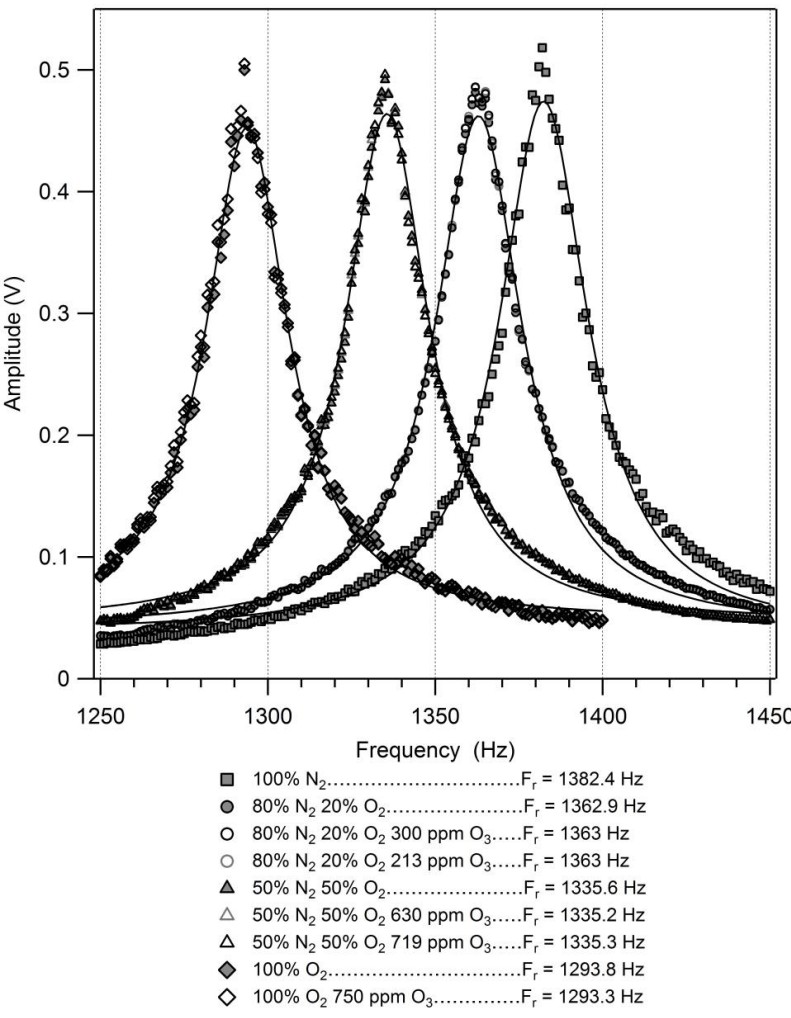


Figure 2: Fast Fourier transform (FFT) resultant power spectra with different carrier gas composition and $O_3$
concentration. $O_3$ was measured downstream to the PAS using the CRD-S assuming $O_3$ $\sigma_{abs}$ of $1.5 \times 10^{-23}$ $cm^2$
molecule$^{-1}$ from Axson et al., (2011).

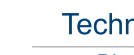

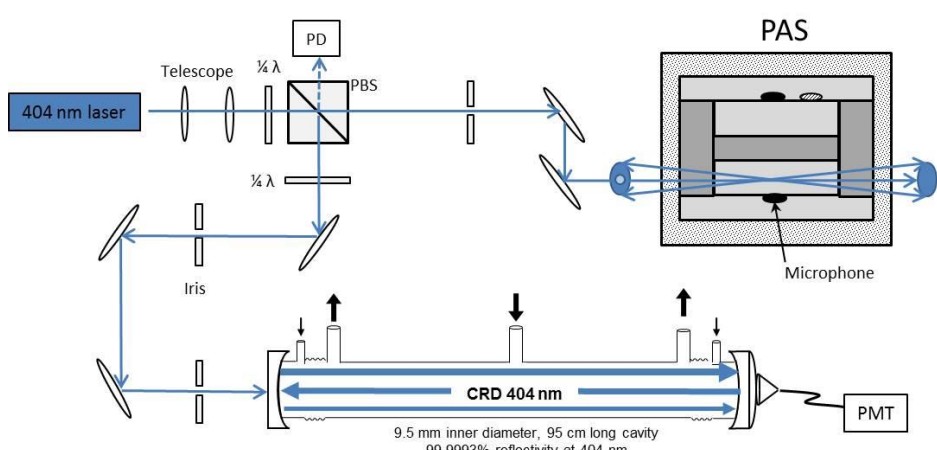


Figure 3: Schematic of the photo-acoustic spectrometer (PAS) coupled to a cavity ring down (CRD) spectrometer (PA-CRD-S). Abbreviations: PBS, polarizing beam splitter; PD, photodiode; PMT, photomultiplier tube. Small black arrows indicate the entrance of the purge flows, and the thinker black arrows the direction of the aerosol flow (Bluvshtein et al., 2016).

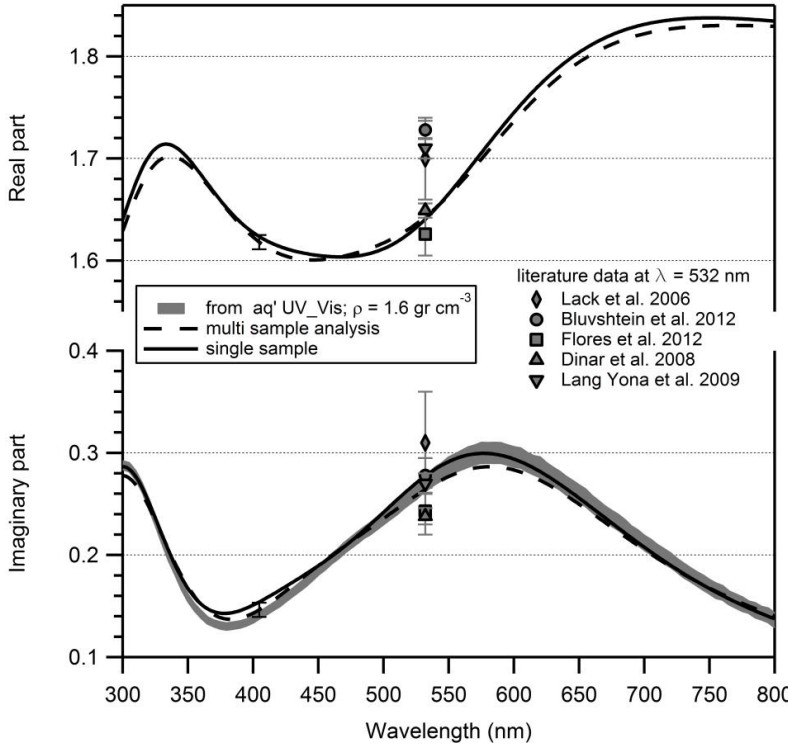

308




Figure 4: Complex RI results of spectroscopic ellipsometry measurements of nigrosin coating on Si oxide. Also shown are results of imaginary part calculated from aqueous UV-Vis measurements based on Sun et al., (2007) with density value of 1.6 gr cm$^{-3}$ (Moteki et al., 2010).

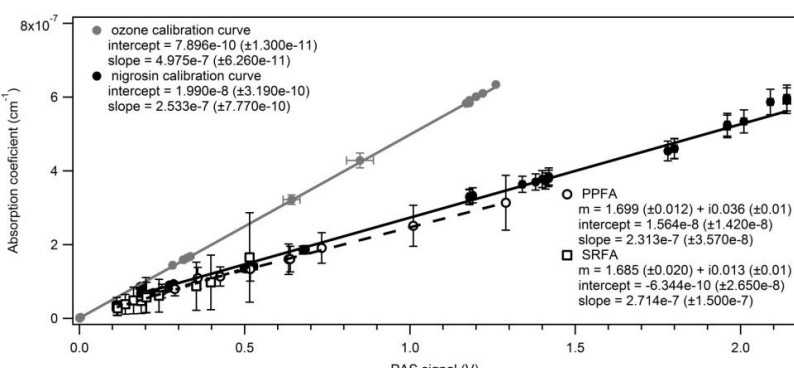

Figure 5: PAS O$_3$ calibration curve and regression (gray), nigrosin calibration curve and regression based on SE analysis (black circles and line) and PPFA and SRFA based on complex RI retrieval from CRD-S measurements and aqueous UV-Vis measurements.

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
