# Peer review of "Calibration of a photoacoustic spectrometer cell using light 1"

_Atmospheric Measurement Techniques, 2016_

## Referee Comment (RC1) · Anonymous Referee #1 · 9 Dec 2016

To reduce uncertainty associated with aerosol impacts on atmospheric radiative transfer and heating, high accuracy measurements of the fundamental absorption properties of atmospheric particles are needed. Photoacoustic spectroscopy (PAS) is increasingly being applied to this problem on account of its potential to offer significantly higher accuracy than traditional techniques. However, the accuracy of the PAS technique depends strongly on the efficacy of its calibration. A common approach used to calibrate aerosol PAS instruments is to use absorption by gaseous ozone. The study of Bluvshtein compares this calibration approach to a new particle-based method. For calibrations at 404nm, it shows disagreement between methods of the order of a factor of 2. These are attributed to problems with the ozone-based approach. Should this

result be generally applicable to PAS systems other than that of the authors, it will have wide ranging impact. However, the paper devotes little attention to understanding the causes for the discrepancies. As such, it leaves significant uncertainty regarding the general applicability of the result to other systems, both in terms of the existence of a bias and its potential magnitude. In summary, this is an important and welcome piece of research that is well suited to AMT, however significant effort should be directed towards understanding and justifying the key results before publication.

Specific comments

- The observed result would be consistent with $NO_2$ contamination in the ozone generation line. Ozone absorbs strongly at 404nm and is also readily photolysed. While the optical power in the CRDS will be insufficient to photolyse $NO_2$, this is not the case for the PAS. Photolysis of $NO_2$ contaminants could lead to the PAS under measuring absorption, as observed. Is this occurring here? There are several ways that the authors could explore this e.g. perform calibrations at different PAS laser powers to change the photolysis rate in the PAS cell. Alternatively, a different analytical technique could be used to quantify $NO_2$ concentration in the calibration flow sample stream. If $NO_2$ absorption is important, then this leads to the question of whether the contamination is unique to this setup or a more general issue.

- Related to the above, the authors describe two methods for generation of ozone – a UV lamp and a corona discharge. It is conceivable that $NO_2$ could be generated from $N_2$ impurities in the $O_2$ carrier gas in the corona discharge. However, a pathway for $NO_2$ generation using the UV lamp method is less clear. Are different ozone calibration slopes observed for these two generation methods?

- One difference between the ozone and nigrosin-based calibrations is the dependence of the former on measurements from the CRDS system. There is no evidence provided to validate the CRDS measurements quantitatively. Presumably the authors would justify the particle CRD measurements based on agreement between the PPFA and SRFA

PAS calibration slopes (which used refractive index values derived from the CRDS) and the nigrosin slope (which used refractive index from ellipsometry measurements). However, this does not provide validation of the gaseous ozone measurements. The cavity RL factor for gas vs aerosol sampling may well be different due to ability of O3 to diffuse into the mirror purge volumes more effectively than aerosol. In the limiting case of O3 filling the full cavity length, this would lead to the CRD over measuring ozone by around 19%. What is the RL factor for gas vs aerosol sampling in the current setup and what is its uncertainty?

- On a related theme, what was the CRD mirror purge flow rate used and was a correction for sample dilution applied in calculation of the CRD ozone concentration?

- Previous studies (e.g. Lack, AST, 2006) have compared ozone-calibrated PAS measurements to Mie calculations for nigrosin and shown agreement to better than 5%. Thus the underlying message of the paper is that the O3 calibration issue is wavelength dependent. Do the authors have the means to verify the result of Lack by repeating measurements at 532nm? This would prove unequivocally that the problem at 404nm is not due to any issues specific to this setup.

- The signal magnitude in the photoacoustic instrument depends on the spatial overlap of the absorbing sample, laser beam and eigenmode pressure distribution within the resonator. Although in theory the eigenmode does not extend beyond the resonator into the buffer volumes, in reality it will. Thus, differences between gas phase and particle calibrations could result from different distributions of aerosol vs calibration gas within the buffer volumes (noting that O3 will diffuse more effectively than particles). In this case, the positioning of sample flow ports on the cells could be important. Where were flow ports positioned on the PAS cells used, and in particular were there differences to Lack et al. (2012)?

- The cavity Q factors for the PAS appear to be significantly lower (by a factor of approx 2) than those reported in Lack et al. (2012) for the same resonators at pressures

around 100kPa. Why is this?

- More detail on the ozone generation system is needed: what flow rates were used and how were flows regulated? What were the plug flow residence times in the gas delivery lines and detection cells? What materials came into contact with the ozone sample? Does changing the flow rate have any impact on the ozone calibration slope? It shouldn't, but such dependency could arise from e.g. changing the timescales for wall loss or changing timescales for production/destruction of contaminant absorbers.

- Where was the CPC positioned during particle calibrations? Presumably after the PAS cell. What were particle losses in the PAS cell and interconnecting tubing? What was the error on the particle count number?

- The authors state that an impactor was used to reduce the concentration of multiply charged particles. How successful was this approach and what was the contribution of multiply charged particles to total absorption in the sample stream? Were larger particles included in Mie calculations used for the nigrosin-based calibrations?

- A lot of technical information is provided regarding the ellipsometry method used to determine the complex refractive index of nigirosin. However, for those unfamiliar with this technique, it is not immediately clear how errors on these measurements are derived. For example, does the structure of the thin layered nigrosin impact its density and refractive index compared to that expected for the bulk. Could this introduce systematic errors in the complex RI determined? Nigrosin density was taken to be 1.6 g cm-3. What is the error on this number and how does it propagate to uncertainty in the complex RI?

- Many of the above questions point to the need for a critical assessment of the errors associated with both the ozone and particle based calibrations. For the latter, the authors need to estimate errors in particle size, size distribution, complex refractive index, number concentrations and propagate these through a Mie routine to see what the impact is on the derived absorption coefficient.

- It is not clear from where the nigrosin used in these studies was sourced. Nigrosin appears to be a complex mixture of components and thus to what extent are its optical properties expected to vary between batches, samples and suppliers? This is important to know if nigrosin is to be adopted widely as a calibration standard.

- Lines 272-280: more information is needed here. How was the system set up in tandem? Was the CRDS sampling before or after the PAS cell (or both) and what flow rates were used? Some quantitative information on results from these tests would be welcome.

Technical corrections

Line 13: reword to avoid starting sentence with 'And'. This doesn't read well.

Line 18: change 'offer' to 'offers'

Line 18: change 'the PAS' to 'photoacoustic instruments' to improve flow.

Line 26: replace 'to the top' with 'at the top'

Line 28: replace 'clouds' with 'cloud'

Line 39: The fact that BC and BrC absorb does not introduce a need to measure absorption. The need is that we need to characterise the absorption properties of BC and BrC with high accuracy to understand impacts on the atmosphere.

Line 70: do the authors mean transmittance measurement here? For example Lack et al compare to CRDS-derived extinction measurements.

Line 76: insert 'phase' to read 'gas phase absorption'

Line 82: correct spelling on 'photolayses'

Lines 95-100: it would be worth stating explicitly here why particle based cals may be harder to implement in field applications e.g. due to the need for particle generation, size selection and counting equipment.

Line 122: would these notches be better described as buffer volumes?

Line 125: replace 'at the background' with 'in the background'

Line 129: replace 'were' with 'where'

Line 193: correct spelling for nigrosin

Line 224: add the RH to which the samples was dried here

---

## Referee Comment (RC2) · N. Wagner (Referee) · 23 Dec 2016

The authors discuss and compare the calibration of a multipass aerosol photoacoustic spectrometer at 404 nm using absorbing aerosol and ozone. The main difficulty with using an absorbing aerosol to calibrate a photoacoustic spectrometer is knowing beforehand the single scattering albedo (or complex index of fraction) of the aerosol. The advance presented in this paper is using an independent measurement of the bulk index of refraction using spectroscopic ellipsometry and a Mie scattering calculation to determine the aerosol absorption coefficient of size-selected aerosol which is then used to calibrate the photoacoustic spectrometer.

The authors note that the calibrations using absorbing aerosol and ozone disagree by

a factor of 2. This is an important and somewhat troubling observation as it may affect several instruments currently in use and the interpretation of historic data form these instruments. As such, it is an important result that should be published in AMT.

The authors leave two important questions unanswered that will be of interested to most readers. First, what is the source of the disagreement between the calibrations? Is the issue specific to the multipass photoacoustic spectrometer, more generally to 404 nm photoacoustic measurement, or does the issue persist with O3 at longer wavelengths across the visible (532nm?)? Is there a non-thermal absorption process in O3 that needs to be better understood?

The authors have likely tried to answer this question and not arrived at a satisfactory answer. While this question does not need to be answered before publication, some more discussion of problems they checked for would be helpful. Here are few experiments that I would like to see the results of. If the authors have already done some of these experiments, including the results would be appropriate. If the instrumentation is available, further checks could be done and discussed.

1) Does the O3 calibration slope vary with laser intensity in the PAS cell? A stable calibration as the PAS laser power is varied would suggest that O3 is not destroyed through any photochemical process. It would also demonstrate that the O3 calibration is not contaminated by NO2 and its photolysis.

2) Can the authors estimate the possible contamination of the O3 produced by the discharge or UV lamp (NO2)? Using either a UV ozone instrument or extinction measurements at variety of wavelengths.

3) Does the O3 calibration curve differ when O3 is generated using the UV lamp compared with the corona discharge?

4) Does the O3 calibration curve agree with the absorbing aerosol calibration at other wavelengths (532 nm, 660 nm) commonly used for photoacoustic spectroscopy?
5) How do these calibrations compare with an extinction-minus-scattering measurements of absorption?

Second, what is the total uncertainty with the nigrosin calibrations? Ozone calibrations are attractive (although possibly flawed due the results reported in this paper) in part because the calibration can be linked to common and accurate UV O3 instruments and the well-measured O3 cross-section in the UV. The nigrosin calibration is susceptible to uncertainty from selection of multiple charge particles in the DMA, uncertainty in the CPC measurement, uncertainty in the nigrosin index of refraction measurement, and uncertainty in applying Mie theory to possibly non-spherical particles. Similar size-selected aerosol calibrations for aerosol mass spectrometers are uncertain at the +/- 35% level. The authors should present an overall uncertainty estimate for the nigrosin calibration before final publication.

Technical comments:

Line 123: Does the PAS instrument here use a single microphone or two (subtracted) as described by Lack et al.?

Line 191: Please state the manufacturer and batch number the nigrosin used here. Its composition can vary from batch-to-batch, and it is not clear how much the absorption (or index of refraction) vary between batches/manufacturers.

Line 193: inconsistent spelling of 'nigrosine'

Line 285: "PAP instrument" should be "PAS instruments"

Figure 2: When describing Fig. 2 the author should clearly state the shift in resonate frequency with increasing O3 concentration is theoretical bias in an theoretical instrument with a much higher-Q acoustic resonator (or calibrations with higher O3 concentrations than used in this paper), and the observed shift of <1 Hz with not affect the O3 calibration slope reported in this paper.

Figure 3: The authors should report their measurements of the nigrosin index of refraction in tabular form so that other groups can apply this information to photoacoustic calibrations at 404 nm and other wavelengths across the visible. Perhaps as supplemental data.

Figure 5: For this O3 calibration, is the O3 generated using the discharge, the lamp, or both?

Figure 5: Uncertainties in the slopes and intercepts are unrealistically small and should include an estimation of systematic errors which are likely larger than the mathematical uncertainty associated with the fit.

---

## Referee Comment (RC3) · A. Petzold (Referee) · 2 Jan 2017

GENERAL COMMENT

The presented work tackles the important experimental question of how to calibrate photoacoustic aerosol instruments for wavelength regimes where no reference gases are available. The authors suggest the use of light absorbing aerosols generated from nebulized light absorbing organic materials which were preselected in size before measurement by means of an electrostatic classifier. The reference absorption coefficient for the calibration is calculated from the size of the aerosol by using Mie theory. The required complex refractive index for the material is determined experimentally by means of spectroscopic ellipsometry. The presented work builds on carefully conducted ex-

perimental studies and deserves publication in AMT after consideration of one major concern which is discussed below.

SPECIFIC REMARKS

1. My major concern refers to the calibration procedure. The method of generating particles of given size and spherical shape, and calculating the absorption coefficient from measured number concentrations by Mie theory is justified and works well for calibrating optical instruments. Here, the additional complexity arises from the fact that the complex index of refraction for the used materials has to be determined separately. The authors demonstrate the robustness of their approach by comparing calibrations with three different materials. They found similar instrument responses for all materials, which is shown in their Figure 5.

On the other hand, they applied the accepted methodology of using ozone as a light-absorbing gas at the selected wavelength of 404 nm. The ozone calibration however produces an instrument response two times higher than the values found for particulate calibration material.

To me, it has to be discussed in more detail which process can cause the differences between the calibration using light absorbing gases or particles. It would be highly beneficial to show simultaneous measurements of light extinction and scattering co-efficients and apply the difference method. A separate proof of the robustness of the calibration by particulate matter combined with Mie theory would be a convincing argument which is not yet given.

2. A full theoretical description of photoacoustic signal generation is provided by Petzold and Niessner (1996), however for an azimuthal resonator. Together with the description of a longitudinal photoacoustic resonator given by Arnott et al. (1999), the authors may investigate potential sources of this discrepancy between the calibration approaches also on a theoretical basis.

MINOR COMMENTS

1. Line 45: The correct reference is Müller et al. (2011).

2. Line 49: correct: ". . . to measure . . ."

3. Line 82: correct: ". . . photolyses . . ."

4. Line 203: I assume the later used acronym SE refers to spectroscopic ellipsometry. If this is the case, it should be introduced here.

5. Line 285: correct "PAS instrument".

6. Figure 5: It should be noted in the y-axis title that the absorption coefficient is obtained from Mie theory.

REFERENCES

Arnott, W. P., Moosmuller, H., Rogers, C. F., Jin, T. F., and Bruch, R.: Photoacoustic spectrometer for measuring light absorption by aerosol: instrument description, Atmos. Environ., 33, 2845-2852, 1999.

Petzold, A., and Niessner, R.: Photoacoustic soot sensor for in-situ black carbon monitoring, Appl. Phys. B, 63, 191-197, 1996.

---

## Author Comment (AC1) · 2 Feb 2017

To reduce uncertainty associated with aerosol impacts on atmospheric radiative transfer and heating, high accuracy measurements of the fundamental absorption properties of atmospheric particles are needed. Photoacoustic spectroscopy (PAS) is increasingly being applied to this problem on account of its potential to offer significantly higher accuracy than traditional techniques. However, the accuracy of the PAS technique depends strongly on the efficacy of its calibration. A common approach used to calibrate aerosol PAS instruments is to use absorption by gaseous ozone. The study of Bluvshtein compares this calibration approach to a new particle-based method. For

calibrations at 404nm, it shows disagreement between methods of the order of a factor of 2. These are attributed to problems with the ozone-based approach. Should this result be generally applicable to PAS systems other than that of the authors, it will have wide ranging impact.

Reply: We thank the Reviewer for the careful reading of the manuscript and his/her supportive comment.

However, the paper devotes little attention to understanding the causes for the discrepancies. As such, it leaves significant uncertainty regarding the general applicability of the result to other systems, both in terms of the existence of a bias and its potential magnitude. In summary, this is an important and welcome piece of research that is well suited to AMT, however significant effort should be directed towards understanding and justifying the key results before publication.

Specific comments: The observed result would be consistent with NO2 contamination in the ozone generation line. Ozone (did you mean NO2?) absorbs strongly at 404nm and is also readily photolysed. While the optical power in the CRDS will be insufficient to photolyse NO2, this is not the case for the PAS. Photolysis of NO2 contaminants could lead to the PAS under measuring absorption, as observed. Is this occurring here?

Reply: NO2 contamination in the corona discharge ozone generation line is possible if H2O and N2 are available. To avoid this possibility, the dry N2 (from a liquid N2) was mixed with the O2/O3 mixture down flow from the corona discharge instrument. Additionally, ozone was generated by two independent sources, with similar results: a constant flow of high purity (99.999%) O2 through the UV lamp O3 generator and a corona discharge ozone generator. These considerations make NO2 contamination in the corona discharge ozone generation line improbable. We addressed this issue in added remarks in the manuscript section "PAS calibration".

There are several ways that the authors could explore this e.g. perform calibrations at

different PAS laser powers to change the photolysis rate in the PAS cell.

Reply: We have performed the O3 calibration at different instrumental configuration and discuss the results in the context of possible NO2 contamination in the supplementary material. We concluded that the effect of such contamination if exist at all is negligible in our system.

Alternatively, a different analytical technique could be used to quantify NO2 concentration in the calibration flow sample stream.

Reply: Following to the above explanation we did not see the need for such measurement.

If NO2 absorption is important, then this leads to the question of whether the contamination is unique to this setup or a more general issue. Related to the above, the authors describe two methods for generation of ozone – a UV lamp and a corona discharge. It is conceivable that NO2 could be generated from N2 impurities in the O2 carrier gas in the corona discharge. However, a pathway for NO2 generation using the UV lamp method is less clear. Are different ozone calibration slopes observed for these two generation methods?

Reply: We show in the supplementary material comparison between different calibration curves performed at different instrumental setup and with using the two O3 generation methods on the same day. Results shows that the calibration slops differ by less than 5%.

One difference between the ozone and nigrosin-based calibrations is the dependence of the former on measurements from the CRDS system. There is no evidence provided to validate the CRDS measurements quantitatively. Presumably the authors would justify the particle CRD measurements based on agreement between the PPFA and SRFA PAS calibration slopes (which used refractive index values derived from the CRDS) and the nigrosin slope (which used refractive index from ellipsometry measurements).

However, this does not provide validation of the gaseous ozone measurements.

Reply: We added a supplementary material section which describes validation of the CRD-S response and data analysis using both gas (N2) and aerosol filled cavity.

The cavity RL factor for gas vs aerosol sampling may well be different due to ability of O3 to diffuse into the mirror purge volumes more effectively than aerosol. In the limiting case of O3 filling the full cavity length, this would lead to the CRD over measuring ozone by around 19%. What is the RL factor for gas vs aerosol sampling in the current setup and what is its uncertainty?

Reply: RL factor calculated from the cavity geometry is 1.187($\pm$ 0.003). The RL factor was additionally calculated by measuring the CRD-S response to 90% N2 10% O2/O3 mixture in a center inlet and side outlets configuration with and without purge flows. The difference between the two calculated values was below 1%. We acknowledged that with side inlet and side outlet CRD-S configuration, the uncertainty in RL may lead to substantial errors in aerosol measurements. For this reason we do not use this CRD-S configuration.

On a related theme, what was the CRD mirror purge flow rate used and was a correction for sample dilution applied in calculation of the CRD ozone concentration?

Reply: The CRD-S purge flows in our system are about 40 cm3min-1 each and dilution correction is not needed because of the center-inlet CRD-S configuration as described in the text.

Previous studies (e.g. Lack, AST, 2006) have compared ozone-calibrated PAS measurements to Mie calculations for nigrosin and shown agreement to better than 5%. Thus the underlying message of the paper is that the O3 calibration issue is wavelength dependent. Do the authors have the means to verify the result of Lack by repeating measurements at 532nm? This would prove unequivocally that the problem at 404nm is not due to any issues specific to this setup.

Reply: Currently a 532 nm PA-CRD-S system is not available and we do not have the ability to perform such validation. Still, for measurements of absorption in short wavelengths this may pose a problem

The signal magnitude in the photoacoustic instrument depends on the spatial overlap of the absorbing sample, laser beam and eigenmode pressure distribution within the resonator. Although in theory the eigenmode does not extend beyond the resonator into the buffer volumes, in reality it will. Thus, differences between gas phase and particle calibrations could result from different distributions of aerosol vs calibration gas within the buffer volumes (noting that O3 will diffuse more effectively than particles). In this case, the positioning of sample flow ports on the cells could be important. Where were flow ports positioned on the PAS cells used, and in particular were there differences to Lack et al. (2012)?

Reply: The PAS cell used in this study is similar to the one described by Lack et al. (2012) and to other instruments used in reported literature (see text for references). We agree with the referee in regards to the potential importance of gas Vs. aerosols diffusion and spatial distribution inside the PAS cell. Unfortunately, we do not have the means to test whether or not this is the cause for the ozone and nigrosin calibration slope discrepancy.

The cavity Q factors for the PAS appear to be significantly lower (by a factor of approx. 2) than those reported in Lack et al. (2012) for the same resonators at pressures around 100kPa. Why is this?

Reply: In our instrument at about 100 kPa and 296 C with synthetic air in the cell the Fr is about 1363 Hz and the FWHM is about 35 Hz leading to Q of 39. Extrapolating from Lack et al. (2012) their cell would have Fr of about 1360Hz and FWHM of about 15 Hz leading to Q of about 90. It seems that the source for difference in Q is in the FWHM parameter (Q/Fr=1/FWHM). We don't have an explanation for this difference but notice that higher value of FWHM means less Fr sensitivity to changes in cell pressure and a

more stable system.

More detail on the ozone generation system is needed: what flow rates were used and how were flows regulated? What were the plug flow residence times in the gas delivery lines and detection cells? What materials came into contact with the ozone sample?

Reply: We have provided additional details regarding ozone generation and flow conditions.

Does changing the flow rate have any impact on the ozone calibration slope? It shouldn't, but such dependency could arise from e.g. changing the timescales for wall loss or changing timescales for production/destruction of contaminant absorbers.

Reply: Changing the gas flow rate does not change the calibration slope. We preformed O3 calibration using the corona discharge generator in a tandem "PAS first" configuration at flow rates of 100, 300, 600, 900 cc min-1 with calibration slopes of 3.967×10-7, 4.047×10-7, 3.985×10-7, 4.020×10-7 cm-1 V-1 respectively.

Where was the CPC positioned during particle calibrations? Presumably after the PAS cell. What were particle losses in the PAS cell and interconnecting tubing? What was the error on the particle count number?

Reply: Additional details of measurement setup flow and tubing was added to the text. Lack et al. (2012) reported less than 1.5% particle loss for sub-micron particles at 1 LPM flow through the PAS cell and inlet tubing. Particle loss was treated as negligible and was not accounted for in these measurements. Errors on particle count number were taken as the standard error on a 120 seconds 1 Hz data set.

The authors state that an impactor was used to reduce the concentration of multiply charged particles. How successful was this approach and what was the contribution of multiply charged particles to total absorption in the sample stream? Were larger particles included in Mie calculations used for the nigrosin-based calibrations?

Reply: Additional details regarding the use of the impactor to reduce the effect of multiply charged particles is provided in the supplementary information. At the nigrosin calibration example given in the main text, multiply charged particles are not included in the Mie routine calculation. However, for the additional analysis presented in figure 5 for SRFA and PPFA, the impactor was not used and the contribution due to multiply charged particles was accounted for both in the complex RI retrieval from CRD-S measurements and in the absorption coefficients that are calculated and presented in figure 5. The agreement between the slope calculated from nigrosin calibration data points and the additional SRFA and PPFA data points implies that the multiply charged particle removal using the impactor worked as well as the multiply charged particles correction used for complex RI retrieval 1-3.

A lot of technical information is provided regarding the ellipsometry method used to determine the complex refractive index of nigirosin. However, for those unfamiliar with this technique, it is not immediately clear how errors on these measurements are derived. For example, does the structure of the thin layered nigrosin impact its density and refractive index compared to that expected for the bulk. Could this introduce systematic errors in the complex RI determined?

Reply: The uncertainty analysis of the SE measurements is described in details in the methodology section. An additional analysis was added to the results section regarding possible effects that air voids may have on the retrieval of the complex RI.

Nigrosin density was taken to be 1.6 g cm-3. What is the error on this number and how does it propagate to uncertainty in the complex RI?

Reply: Nigrosin density of 1.6 gr cm-3 was taken from Moteki et al. 4 where it is reported without uncertainty. However the experimental setup used in this paper (DMA-APM; differential mobility analyzer coupled to an aerosol particle mass analyzer) was reported in Mcmurry et al. 5 with an average uncertainty of 5%. This was used to re-calculate the range of imaginary part of the complex RI of nigrosin from the UV-Vis absorption measurements of diluted aqueous solution.

Many of the above questions point to the need for a critical assessment of the errors associated with both the ozone and particle based calibrations. For the latter, the authors need to estimate errors in: - particle size, - size distribution, - complex refractive index, - number concentrations and propagate these through a Mie routine to see what the impact is on the derived absorption coefficient.

Reply: The uncertainties of the complex refractive index retrieved from the ellipsometer measurement and the precision of the number concentration (as standard error of 120 continuous samples) were propagated through the Mie routine as described in the text. The uncertainties on the size distribution was added to the analysis. Table 1 lists all components of uncertainty propagated through the Mie routine.

It is not clear from where the nigrosin used in these studies was sourced.

Reply: Nigrosin was purchased from Sigma-Aldrich (batch number: 14828BD). This information was added in the text.

Nigrosin appears to be a complex mixture of components and thus to what extent are its optical properties expected to vary between batches, samples and suppliers? This is important to know if nigrosin is to be adopted widely as a calibration standard.

Reply: Unfortunately, this is a question neither I, nor the supplier (Sigma-Aldrich) can answer. In order to test this issue, I would recommend future users to measure absorption of diluted aqueous solutions and compare calculated imaginary part to results reported in the manuscript or perform ellipsometer measurements using their stock material.

Lines 272-280: more information is needed here. How was the system set up in tandem? Was the CRDS sampling before or after the PAS cell (or both) and what flow rates were used? Some quantitative information on results from these tests would be welcome.

Reply: Additional information regarding the tandem and parallel system configurations

was added to the main text and quantitative comparison between the different configurations is now presented in the supplementary information.

Technical corrections:

Line 13: reword to avoid starting sentence with 'And'. This doesn't read well.

Reply: Was rephrased

Line 18: change 'offer' to 'offers'

Reply: Was rephrased

Line 18: change 'the PAS' to 'photoacoustic instruments' to improve flow.

Reply: Was rephrased

Line 26: replace 'to the top' with 'at the top'

Reply: Was rephrased

Line 28: replace 'clouds' with 'cloud'

Reply: Was rephrased

Line 39: The fact that BC and BrC absorb does not introduce a need to measure absorption. The need is that we need to characterise the absorption properties of BC and BrC with high accuracy to understand impacts on the atmosphere.

Reply: Thank you. The text was modified: "Light absorption properties of BrC and its mixing with absorbing and non-absorbing aerosol components introduce a need for sensitive and accurate direct measurement of light absorbing aerosols which is still very challenging." Was rephrased to: "Light absorption properties of BrC and its mixing with absorbing and non-absorbing aerosol components introduce a need for sensitive and accurate measurements of light absorbing aerosols in order to improve our understanding of the impacts of absorbing aerosols on climate."

Line 70: do the authors mean transmittance measurement here? For example Lack et al compare to CRDS-derived extinction measurements.

Reply: We did mean transmittance. CRD-S extinction measurements are essentially measurements of transmition.

Line 76: insert 'phase' to read 'gas phase absorption'

Reply: Was rephrased

Line 82: correct spelling on 'photolayses'

Reply: Was corrected

Lines 95-100: it would be worth stating explicitly here why particle based cals may be harder to implement in field applications e.g. due to the need for particle generation, size selection and counting equipment.

Reply: The sentence was rephrased to: "An alternative calibration method is to use a standard aerosol with well-known absorption properties. PAS calibration using size selected light absorbing particles requires a standard material with accurate information of its complex refractive index at the instrument's wavelength, which is not widely available. This procedure is also time consuming in comparison to the use of a light absorbing gas and may be more difficult to implement on field and aircraft applications due to the need for aerosols generation and size selection equipment."

Line 122: would these notches be better described as buffer volumes?

Reply: The two buffer volumes act as acoustic notch filters to reduce acoustic noise at the cell resonant frequency. The term acoustic notches was used by Lack et al. 6 to describe this instrument.

Line 125: replace 'at the background' with 'in the background'

Reply: Was rephrased

Line 129: replace 'were' with 'where'

Reply: Was corrected

Line 193: correct spelling for nigrosin

Reply: Was corrected

Line 224: add the RH to which the samples was dried here

Reply: (RH < 10%) was added in the text

Please also note the supplement to this comment:
http://www.atmos-meas-tech-discuss.net/amt-2016-323/amt-2016-323-AC1-
supplement.pdf

---

## Author Comment (AC2) · 2 Feb 2017

N. Wagner (Referee#2): The authors discuss and compare the calibration of a multipass aerosol photoacoustic spectrometer at 404 nm using absorbing aerosol and ozone. The main difficulty with using an absorbing aerosol to calibrate a photoacoustic spectrometer is knowing beforehand the single scattering albedo (or complex index of fraction) of the aerosol. The advance presented in this paper is using an independent measurement of the bulk index of refraction using spectroscopic ellipsometry and a Mie scattering calculation to determine the aerosol absorption coefficient of size-selected aerosol which is then used to calibrate the photoacoustic spectrometer. The authors note that the calibrations using absorbing aerosol and ozone disagree by a factor of 2.

This is an important and somewhat troubling observation as it may affect several instruments currently in use and the interpretation of historic data form these instruments. As such, it is an important result that should be published in AMT.

Reply: We thank Dr. Wagner for the careful reading of the manuscript and for these supporting statements.

The authors leave two important questions unanswered that will be of interested to most readers. First, what is the source of the disagreement between the calibrations? Is the issue specific to the multipass photoacoustic spectrometer, more generally to 404 nm photoacoustic measurement, or does the issue persist with O3 at longer wavelengths across the visible (532nm?)? Is there a non-thermal absorption process in O3 that needs to be better understood?

Reply: We currently cannot answer this question because we only have the PAS-CRD-S system at 404 nm wavelength. However, in our understanding there is no known non-thermal absorption process for O3 at this wavelength.

The authors have likely tried to answer this question and not arrived at a satisfactory answer. While this question does not need to be answered before publication, some more discussion of problems they checked for would be helpful. Here are few experiments that I would like to see the results of. If the authors have already done some of these experiments, including the results would be appropriate. If the instrumentation is available, further checks could be done and discussed. 1) Does the O3 calibration slope vary with laser intensity in the PAS cell? A stable calibration as the PAS laser power is varied would suggest that O3 is not destroyed through any photochemical process. It would also demonstrate that the O3 calibration is not contaminated by NO2 and its photolysis.

Reply: Additional information related to this issue was added to the supplementary information. We show that the PAS signal to O3 light absorption is linear with laser power.

2) Can the authors estimate the possible contamination of the O3 produced by the discharge or UV lamp (NO2)? Using either a UV ozone instrument or extinction measurements at variety of wavelengths.

Reply: Currently we do not have sensitive enough NO2 measurement capability. We took measures and tests to ensure that NO2 or other contamination if exist is negligible. A discussion on this issue was added to the supplementary information.

3) Does the O3 calibration curve differ when O3 is generated using the UV lamp compared with the corona discharge?

Reply: We show in the supplementary material comparison between different calibration curves performed at different instrumental setup and with using the two O3 generation methods on the same day. Results shows that the calibration slops differ by less than 5%.

4) Does the O3 calibration curve agree with the absorbing aerosol calibration at other wavelengths (532 nm, 660 nm) commonly used for photoacoustic spectroscopy?

Reply: Since we currently only have a PA-CRD-S system at 404 nm we cannot answer this question. However, due to the need to measure in the short wavelength spectral range, it is important to understand processes in these wavelengths where potential problems may arise.

5) How do these calibrations compare with an extinction-minus-scattering measurements of absorption?

Reply: We currently do not have nephelometer capable of measuring scattering at 404 nm wavelength so we cannot address this question as well.

Second, what is the total uncertainty with the nigrosin calibrations? Ozone calibrations are attractive (although possibly flawed due the results reported in this paper) in part because the calibration can be linked to common and accurate UV O3 instruments and the well-measured O3 cross-section in the UV. The nigrosin calibration is susceptible

to uncertainty from selection of multiple charge particles in the DMA, uncertainty in the CPC measurement, uncertainty in the nigrosin index of refraction measurement, and uncertainty in applying Mie theory to possibly non-spherical particles. Similar size selected aerosol calibrations for aerosol mass spectrometers are uncertain at the +/- 35% level. The authors should present an overall uncertainty estimate for the nigrosin calibration before final publication.

Reply: Additional discussion regarding the error propagation through the Mie routine and its implication to the PAS calibration curve was added to the main text. Table 1 lists all components of uncertainty propagated used.

Technical comments: Line 123: Does the PAS instrument here use a single microphone or two (subtracted) as described by Lack et al.?

Reply: Two microphones are used in this instrument as described in Lack et al. 6 and in the following line in the text.

Line 191: Please state the manufacturer and batch number the nigrosin used here. Its composition can vary from batch-to-batch, and it is not clear how much the absorption (or index of refraction) vary between batches/manufacturers.

Reply: Nigrosin was purchased from Sigma-Aldrich (batch number: 14828BD). This information was added in the text.

Line 193: inconsistent spelling of 'nigrosine'

Reply: Was changed to 'nigrosin'

Line 285: "PAP instrument" should be "PAS instruments"

Reply: Was corrected

Figure 2: When describing Fig. 2 the author should clearly state the shift in resonate frequency with increasing O3 concentration is theoretical bias in an theoretical instrument with a much higher-Q acoustic resonator (or calibrations with higher O3 concentrations than used in this paper), and the observed shift of <1 Hz with not affect the O3 calibration slope reported in this paper.

Reply: The following section was added in the discussion of figure 2: "Although figure 2 demonstrates that no change in Fr could be detected in our instrument with increasing O3 concentration of up to 750 ppm, such a shift due to gas composition change is possible in higher Q acoustic resonator and with higher O3 concentration."

Figure 3: The authors should report their measurements of the nigrosin index of refraction in tabular form so that other groups can apply this information to photoacoustic calibrations at 404 nm and other wavelengths across the visible. Perhaps as supplemental data.

Reply: The spectroscopic ellipsometry data was added in table S1 of the supplementary material.

Figure 5: For this O3 calibration, is the O3 generated using the discharge, the lamp, or both?

Reply: The O3 used in the calibration curve presented in figure 5 was prepared using the corona discharge O3 generator. This is clarified in the main text with more detailed description of the instrumental setup.

Figure 5: Uncertainties in the slopes and intercepts are unrealistically small and should include an estimation of systematic errors which are likely larger than the mathematical uncertainty associated with the fit.

Reply: Additional sources of error were used and propagated through the Mie routine. These increase the calculated uncertainty of the PAS calibration curve. Figure 5 and the related parts in the text were changed accordingly. However, the results and conclusions remain robust.

Please also note the supplement to this comment:

http://www.atmos-meas-tech-discuss.net/amt-2016-323/amt-2016-323-AC2-supplement.pdf

---

## Author Comment (AC3) · 2 Feb 2017

A. Petzold (Referee#3): The presented work tackles the important experimental question of how to calibrate photoacoustic aerosol instruments for wavelength regimes where no reference gases are available. The authors suggest the use of light absorbing aerosols generated from nebulized light absorbing organic materials which were preselected in size before measurement by means of an electrostatic classifier. The reference absorption coefficient for the calibration is calculated from the size of the aerosol by using Mie theory. The required complex refractive index for the material is determined experimentally by means of spectroscopic ellipsometry. The presented work builds on carefully conducted experimental studies and deserves publication in
AMT after consideration of one major concern which is discussed below.

Replay: We thank Dr. Petzold for the time and thoughtful comments to the manuscript.

SPECIFIC REMARKS 1. My major concern refers to the calibration procedure. The method of generating particles of given size and spherical shape, and calculating the absorption coefficient from measured number concentrations by Mie theory is justified and works well for calibrating optical instruments. Here, the additional complexity arises from the fact that the complex index of refraction for the used materials has to be determined separately. The authors demonstrate the robustness of their approach by comparing calibrations with three different materials. They found similar instrument responses for all materials, which is shown in their Figure 5. On the other hand, they applied the accepted methodology of using ozone as a light absorbing gas at the selected wavelength of 404 nm. The ozone calibration however produces an instrument response two times higher than the values found for particulate calibration material. To me, it has to be discussed in more detail which process can cause the differences between the calibration using light absorbing gases or particles.

Replay: We appreciate this comment and have done our best to trace the source of the differences. We describe the study and all the steps we conducted in order to understand this and we welcome following work by our colleagues who use same instrumentation.

It would be highly beneficial to show simultaneous measurements of light extinction and scattering coefficients and apply the difference method.

Replay: Unfortunately, we do not have a nephelometer at 404 nm so the extinction minus scattering validation of the absorption coefficient could not be applied here.

A separate proof of the robustness of the calibration by particulate matter combined with Mie theory would be a convincing argument which is not yet given.

Replay: We show for three different light absorbing organic materials that O3 calibra-

tion of the PA-CRD-S system results in significant over estimation of their absorption coefficient. While aerosol calibration adds technical complexity and uncertainty compared with gas phase calibration we are certain that sufficient evidence was presented to show that significant bias in light absorption properties may be avoided using the proposed methodology.

2. A full theoretical description of photoacoustic signal generation is provided by Petzold and Niessner (1996), however for an azimuthal resonator. Together with the description of a longitudinal photoacoustic resonator given by Arnott et al. (1999), the authors may investigate potential sources of this discrepancy between the calibration approaches also on a theoretical basis.

Replay: The proposed papers were considered and are cited in the main text. However, non of them described results which may be related to what we results described in our manuscript.

MINOR COMMENTS 1. Line 45: The correct reference is Müller et al. (2011).

Replay: Corrected 2. Line 49: correct: ": : : to measure : : :"

Replay: Corrected

3. Line 82: correct: ": : : photolyses : : :"

Replay: Corrected

4. Line 203: I assume the later used acronym SE refers to spectroscopic ellipsometry. If this is the case, it should be introduced here.

Replay: The acronym SE was introduced in this line

5. Line 285: correct "PAS instrument".

Replay: Was corrected

6. Figure 5: It should be noted in the y-axis title that the absorption coefficient is

obtained from Mie theory.

Replay: The following line was added to the figure caption: "Absorption coefficient for the nigrosin, SRFA and PPFA were calculated using Mie theory routine."

Please also note the supplement to this comment:
http://www.atmos-meas-tech-discuss.net/amt-2016-323/amt-2016-323-AC3-supplement.pdf